# The N-Terminus Makes the Difference: Impact of Genotype-Specific Disparities in the N-Terminal Part of The Hepatitis B Virus Large Surface Protein on Morphogenesis of Viral and Subviral Particles

**DOI:** 10.3390/cells9081898

**Published:** 2020-08-13

**Authors:** Bingfu Jiang, Xingjian Wen, Qingyan Wu, Daniela Bender, Gert Carra, Michael Basic, Alica Kubesch, Kai-Henrik Peiffer, Klaus Boller, Eberhard Hildt

**Affiliations:** 1Division of Virology, Paul-Ehrlich-Institut, D-63225 Langen, Germany; Bingfu.Jiang@pei.de (B.J.); Xingjian.Wen@pei.de (X.W.); Qingyan.Wu@pei.de (Q.W.); daniela.bender@pei.de (D.B.); gert.carra@pei.de (G.C.); Michael.Basic@pei.de (M.B.); Alica.Kubesch@kgu.de (A.K.); Kai-Henrik.Peiffer@kgu.de (K.-H.P.); 2Department of Gastroenterology and Hepatology, J. W. Goethe University, D-60590 Frankfurt, Germany; 3Department of Immunology, Paul-Ehrlich-Institut, D-63225 Langen, Germany; klaus.boller@pei.de; 4TTU Hepatitis, German Center for Infection Research (DZIF), 38124 Braunschweig, Germany

**Keywords:** HBV surface protein, PreS1 deletion, HBV filaments, HBV genotypes, morphogenesis

## Abstract

The N-terminus of the hepatitis B virus (HBV) large surface protein (LHB) differs with respect to genotypes. Compared to the amino terminus of genotype (Gt)D, in GtA, GtB and GtC, an additional identical 11 amino acids (aa) are found, while GtE and GtG share another similar 10 aa. Variants of GtB and GtC affecting this N-terminal part are associated with hepatoma formation. Deletion of these amino-terminal 11 aa in GtA reduces the amount of LHBs and changes subcellular accumulation (GtA-like pattern) to a dispersed distribution (GtD-like pattern). Vice versa, the fusion of the GtA-derived N-terminal 11 aa to GtD causes a GtA-like phenotype. However, insertion of the corresponding GtE-derived 10 aa to GtD has no effect. Deletion of these 11aa decreases filament size while neither the number of released viral genomes nor virion size and infectivity are affected. A negative regulatory element (aa 2–8) and a dominant positive regulatory element (aa 9–11) affecting the amount of LHBs were identified. The fusion of this motif to eGFP revealed that the effect on protein amount and subcellular distribution is not restricted to LHBs. These data identify a novel region in the N-terminus of LHBs affecting the amount and subcellular distribution of LHBs and identify release-promoting and -inhibiting aa residues within this motive.

## 1. Introduction

The human hepatitis B virus (HBV) belongs to the family of *Hepadnaviridae*, a group of small hepatotropic DNA viruses. The HBV virion is a spherical particle, 42 nm in diameter, consisting of an inner 30-nm icosahedral nucleocapsid and an outer envelope. The nucleocapsid is mainly formed by the core protein (HBcAg) and harbors the viral genome and polymerase. The polymerase contains a terminal protein domain followed by a spacer domain, a reverse transcriptase domain and an RNase H domain. The envelope is composed of cellular lipid and HBV surface antigen (HBsAg), which contains three different surface proteins: the large HBV surface protein (LHB), the middle surface protein (MHB) and the small surface protein (SHB), the major HBsAg component. The surface proteins share an S domain with 226 amino acids (aa), which is the major component of SHBs. MHBs contains an additional N-terminal PreS2 domain with 55 aa while LHBs consists of an additional PreS1-PreS2 domain. The length of the N-terminal PreS1-domain is genotype (Gt)-dependent between 108 or 118 or 119 aa. The N-terminus of LHBs harbours a myristoylation motif. Myristoylation of LHBs is crucial for HBV infectivity [1]. The envelope proteins are not only constitutive components of the surface envelope of viral particles, but also assemble into subviral particles with the shape of spheres and filaments lacking any capsid and viral DNA [2]. SHBs, the predominant part of these subviral particles, can assemble to 22-nm spherical particles. The incorporation of larger amounts of LHBs in these subviral particles results in the formation of filaments with the same diameter but variable lengths [3,4]. Furthermore, LHBs are involved in viral entry [5] and viral envelopment [6] and promote liver carcinogenesis [7,8]. In addition, there is evidence for the formation of fully enveloped capsids, but not genome free viral particles [9].

Nine HBV genotypes A to I have been identified [10]. The HBV genotypes differ with respect to their endemicity, transmission mode and their pathogenicity [11]. While GtA and GtD are dominant in Europe, GtB and GtC are prevalent in Asia [12]. Sequence analysis reveals that the N-terminal part of the PreS1-domain differs between the genotypes. As compared to the N-terminus of the PreS1 domain of GtD, an N-terminal extension formed by identical 11 amino acids (aa) with an initiating methionine is found in the case of GtA, GtB and GtC, while in the case of GtE and GtG, a different peptide encompassing 10 aa with an initiating methionine forms the N-terminus (Figure 1). In accordance with this, the mutant GtD(+)10aa (GtE) lacks a methionine at position +12.

Variants of HBV GtB and GtC with deletions in the PreS1 start codon and a consequent generation of GtD-like LHBs lacking N-terminal 11 aa have been found to be related to liver cirrhosis and hepatocellular carcinoma (HCC) in Asian patients suffering from chronic infection [13,14,15,16]. However, the impact of these very N-terminal amino acids on the amount and localization of LHBs, the viral life cycle and virus-associated pathogenesis is still unclear.

## 2. Materials and Methods

### 2.1. Study Population

In total, 115 participants (97 White, 7 Asians, 11 African-Americans) of the Albatros trial (listed at Clinical.Trial.gov: NCT01090531) with HBeAg-negative chronic HBV GtA infection (HBeAg-negative, normal transaminases, HBV DNA < 20,000 IU/mL) were included in the analysis. Main inclusion and exclusion criteria of the Albatros trial were described recently [17].

### 2.2. Cells and Transient Transfection

The human hepatocarcinoma cell line Huh7.5 was grown in DMEM (Lonza, Belgium) and supplemented with 10% FCS, 0.1 U/mL penicillin, 100 μg/mL streptomycin and 2 mM L-glutamine (DMEM complete). Huh7.5 cells were transfected using linear polyethyleneimine (PEI) (Polysciences, Inc., Hirschberg, Germany) as described [18].

### 2.3. Antibodies and Chemicals

The monoclonal MA18/7 anti-PreS1 domain of LHBs was kindly provided by Dr. Dieter Glebe, Giessen [3,19]. The polyclonal rabbit against denatured HBV core K46 [20] was a gift from Dr. Reinhild Prange, Department of Medical Microbiology and Hygiene, Johannes Gutenberg-Universität Mainz, Mainz, Germany. Polyclonal rabbit anti-HBV core B0586 was purchased from Dako, Denmark. Anti-beta-Actin was ordered from Sigma-Aldrich (St. Louis, MO, USA). For immunofluorescence staining, Alexa Fluor 488-conjugated (Invitrogen), Cy3- and Cy5-conjugated secondary antibodies (Jackson Immuno Research Laboratories, Inc., Ely, UK) were used. For quantification of HBcAg, HBsAg and HBeAg, a QuickTiterTM Hepatitis B Core Antigen (HBcAg) ELISA Kit (Cell Biolabs, INC, San Diego, CA, USA), and an Enzygnost^®^ HBsAg 6.0 and HBeAg (ELISA; Siemens, Munich, Germany) were ordered. The antobodies were generated by Seramun Institute (Heidesee, Germany). The ethic approval umber is from the RP Potsdam: 2347-A-15-1-2018.

### 2.4. Plasmids

A plasmid GtA carrying 1.1-fold CMV-driven genomes of genotype A2 was used. This context was chosen as a variety of deletion mutants have been established in previous studies of the lab. Plasmids GtD and GtE (E2) containing 1.1-fold CMV-driven genomes were kindly provided by Dr. Stephan Urban, Heidelberg. The mutants harboring either deletions, insertions or point mutations were generated by overlapping PCR using primers listed in the Appendix A.

### 2.5. SDS-PAGE and Western Blot Analysis

SDS-PAGE and Western blot analysis were performed according to standard procedures [21]. Detection of bound secondary antibody was performed either by ECL using Immobilon Western Chemiluminescent HRP Substrate (Merck Millipore, Darmstadt, Germany) or by LI-COR Odyssey Infrared Imager (Biosciences, Lincoln, NE, USA) using secondary antibodies from LI-COR.

### 2.6. Indirect Immunofluorescence Analysis

Fixation and staining were performed as described [22]. Immunofluorescence staining was analyzed using a confocal laser scanning microscope (CLSM 510meta Carl Zeiss, Jena, Germany).

### 2.7. Isolation of RcDNA and PreS1 Direct Sequencing

As mentioned before [17], viral DNA was extracted from 200 µL of serum using the QIAamp DNA Blood Mini Kit (Qiagen, Hilden, Germany) and QIAamp UltraSens Virus Kit (Qiagen, Hilden, Germany) according to the manufacturer’s protocol. The corresponding DNA was subjected to sequencing PCR according to the manufacturer’s instructions (BigDyeDeoxy Terminators; Applied Biosystems, Waltham, MA, USA). DNA was sequenced on a 3130 × l Genetic Analyser (Applied Biosystems, Waltham, MA, USA). A sensitivity level of about 15–20% was assumed.

### 2.8. Rate-Zonal Sedimentation

HBsAg-containing particles after PEG precipitation were dialyzed against STE buffer (0.1 M NaCl, 0.01 M Tris-HCl, 0.001 M EDTA, pH 8.0) and then layered on preformed gradients composed of 2 mL 10, 15, 20, 25 and 30% sucrose (*w/w*) in STE. Ultracentrifugation was performed in a Beckman SW41 Ti rotor at 40,000 rpm for 4 h at 4 °C [23,24]. Fractions of 0.5 mL were collected from above, and their refractive indexes were measured to guarantee the fractionation.

### 2.9. Electron Microscopy

HBsAg-containing particles were precipitated by 10% PEG at 10,000 rpm for 30 min. After solubilization the particles were subsequently either enriched by ultracentrifugation in a Beckman SW60 Ti rotor at 42,000 rpm for 2.5 h in 10% sucrose (*w/w*) at 4 °C or partially purified using rate-zonal sedimentation as described above [23,24]. For negative staining, freshly glow-discharged carbon-coated nickel grids were incubated with sucrose gradient-derived samples for 3 min and washed with distilled water twice. The grids were negatively stained with 2% aqueous phosphotungstic acid for 10 s and examined in a Zeiss EM-109 transmission electron microscope equipped with a TRS 1K camera. The size of filaments and viral particles was measured using ITEM 5 software (EMSIS, Münster, Germany).

### 2.10. HBV Infection of Differentiated HepaRG Cells

HBV infection experiments were performed as described previously [25]. In brief, infectious inocula were prepared from cell culture supernatants HuH7.5 cells transfected with the genome GtA, GtD and the mutants by precipitating viral particles in the presence of 6% polyethylene glycol (PEG) 8000. Infections were conducted in differentiated HepaRG cells at multiplicities of infection (MOI) of 200 and 1600 in the presence of 4% PEG 8000 for 16 h. After infection, the cells were washed three times with the culture medium, and grown in the differentiation medium containing 2% DMSO, 5 μg/mL insulin and 5 × 10^−5^ M hydrocortisone hemisuccinate. Culture supernatants were harvested at the indicated time points and analyzed by HBsAg and HBeAg ELISA.

### 2.11. Statistical Analysis

The significance of results was analyzed by two-tailed t-test using GraphPad Prism version 7.00. Error bars represent standard deviation. *, *p* < 0.05; **, *p* < 0.01; ***, *p* < 0.001.

## 3. Results

### 3.1. Prevalence of PreS1 Start Codon Deletion Variants in European Patients Infected Chronically by GtA

Variants with deletion of the PreS1 start codon have been isolated frequently in Asian patients suffering from chronic infection with GtB and GtC. Moreover, these variants are most frequently isolated from Asian patients with liver cirrhosis and hepatocellular carcinoma [13,14,15]. To investigate whether analogous GtA variants with deletion of the PreS1 start codon could also be found in a large European cohort with HBeAg negative HBV chronically infected patients [17] of isolates from patients chronically infected with HBV-GtA, were analyzed. Variants of GtA with deletion of five (serum-1) or 6 (serum-2) amino acids starting from the initiating methionine were found in 2 of 115 cases (Table 1). The deletion was further confirmed by Western blot using the PreS1-specific monoclonal Ma18/7. Due to the deletion of the first start codon, the translation of LHBs started from the following methionine (as in GtD) and a shift of LHBs to a lower molecular weight was observed (Figure 2A). No significant difference in the level of HBV-specific DNA and HBsAg could be observed between patients replicating the variants or the genome with the intact PreS1 start codon (Table 1). These data indicate that deletion of the first start codon of the PreS1 domain that leads to a N-terminally truncated LHBs is not restricted to GtB and GtC.

### 3.2. The N-Terminal 11aa in the PreS1 Domain of GtA Have an Impact on the Amount of Extra- and Intracellular LHBs

The loss of the first methionine in the PreS1 domain could also be observed for variants of GtA, GtB and GtC, which results in the formation of LHBs lacking the N-terminal 11 aa as in case of GtD. In contrast to the genotypes A-C, GtE and GtG share an amino-terminal peptide that encompasses 10 aa and differs from the sequence of the corresponding peptide of genotypes A–C.

To study the impact of these N-terminal 11 aa of genotypes A–C and of the corresponding 10 aa of genotypes E and G on the amount and distribution of LHBs and on the viral life cycle, the following mutants were generated: a GtA mutant (GtA_∆11aa) lacking these N-terminal 11 aa in the PreS1 domain and two GtD mutants (GtD_(+)11aa(GtA) and GtD_(+)10aa(GtE)) containing either GtA-derived 11 aa or GtE-derived 10 aa in the N-terminal part of the PreS1 domain, respectively (Figure 1). It should be mentioned that apart from the PreS1 domain, the backbones of GtC and GtB differ significantly from GtA and GtD, which should be considered. This accounts for the comparative analysis of GtE and GtG as well. Deletion of the first start codon in the N-terminus of the PreS1 domain of GtA shifts the translation initiation of LHBs to the second methionine in the PreS1 domain. This leads to LHBs with lower molecular weight, while fusion of 10 aa or 11 aa to the N-terminus of the PreS1 domain of GtD shifts the initiation of LHBs to the inserted methionine and generated LHBs with higher molecular weight (Figure 2B). Interestingly, as compared to GtA, the deletion of 11 aa strongly reduced the amount of intracellular and extracellular LHBs. Vice versa, the fusion of GtA-derived 11 aa to the N-terminus of the GtD PreS1 domain (GtD_(+)11aa(GtA) significantly increased the amount of intracellular and extracellular LHBs in comparison to GtD (Figure 2B–D). In contrast, the fusion of the GtE-derived 10 aa to the N-terminus of GtD did not affect the amount of intra- and extracellular LHBs, arguing against a simple spacer function of this region (Figure 2B–D). Further analyses revealed that the half-life of LHBs (GtD) was not affected by fusion of the GtA-derived 11 aa but was shortened by fusion of the GtE-derived peptide (Appendix A) To further analyze whether the deletion or insertion of these N-terminal PreS1 residues affected the production and secretion of core protein and HBsAg, the intracellular and extracellular amounts of core and HBsAg were analyzed by Western blot and ELISA (Figure 2B,E–J). No significant difference in the amounts of intracellular and extracellular core and HBsAg could be observed between GtA and GtA_∆11aa, GtD and GtD_(+)11aa (GtA). This indicates that the changes in LHB production and release are caused by GtA-derived 11 aa specifically. For an unknown reason, a slight but significant increase of extracellular HBsAg (Figure 2J) and intracellular core (Figure 2F) was observed if the GtE-derived 10 aa were fused to the N-terminus of the PreS1 domain of GtD (GtD_(+)10aa(GtE)). These data indicate that in the case of GtA, the amino-terminal 11 aa of the PreS1 domain specifically affects the amount of LHBs but has no general effect on HBsAg and core protein.

### 3.3. The Aminoterminal 11 aa in the PreS1-Domain of GtA Affect the Subcellular Distribution of LHBs

As the presence of GtA-derived 11 aa is associated with a higher amount of intracellular LHBs, its effect on the subcellular distribution of LHBs was analyzed by confocal immunofluorescence microscopy using the LHB-specific MA18/7. As shown in Figure 3, in GtA-expressing cells, the LHB-specific signal was distributed throughout the cytoplasm with significant dot-like accumulation in the perinuclear region (GtA pattern). In cells expressing the GtD genome, a more homogenous distribution of LHBs all over the cytoplasm was found (GtD pattern). Deletion of the N-terminal 11 aa in GtA changed the distribution pattern of LHBs from the GtA pattern to the GtD-like pattern. The fusion of the GtA-derived 11 aa to the N-terminus of PreS1-domain in GtD switched the subcellular distribution of LHBs from the GtD pattern to the almost GtA-like pattern. However, no significant alteration in the subcellular distribution of LHBs could be observed between cells overexpressing the genomes GtD and GtD_(+)10aa(GtE). In summary, the GtA-derived 11 aa not only affect the amount of intracellular LHBs, but also change the subcellular distribution of LHBs.

### 3.4. The N-Terminal Part of the PreS1-Domain of GtA Has an Impact on the Morphology of Filaments

The amino-terminal amino acids of LHBs affect the distribution and amount of LHBs. It is unclear whether they additionally have an impact on the morphology of LHB-containing filaments and viral particles. To investigate this, the enriched extracellular HBsAg derived from cells expressing GtA, GtA_∆11aa, GtD, GtD_(+)10aa(GtE) and GtD_(+)11aa(GtA) was analyzed by rate-zonal sedimentation followed by Western blot (Figure 4A,B). In the case of GtA LHBs was detected with two major peaks in fractions 20 and 24 of the gradient, while for GtA_∆11aa-derived LHBs the peaks were detected in fractions 17 and 24. So, deletion of the N-terminal 11 aa in the PreS1 domain of GtA shifted the first LHB-rich signal to lower density. In the case of GtD, with the two major peaks were found in fractions 15 and 24, whereas for GtD_(+)11aa(GtA) the LHBs peaks were detected in fractions 21 and 24. Insertion of the amino-terminal GtA-derived 11 aa shifted the first LHB-rich peak to higher density. In the case of GtD(+)10aa(GtE), fusion of 10 aa to the N-terminal PreS1 also shifted the first LHB-rich signal to higher density (fraction 18) as compared to GtD. The first peak represents the LHB-containing filaments [23,26]. The shift of the first LHBs peak for the different variants suggests that the morphology of filaments could be affected. To experimentally analyze this, the respective fractions were analyzed by electron microscopy (Figure 4C,D). Lots of filaments with a length of 99.20 nm and diameter of 24.17 nm on average could be observed in GtA-derived fraction 20. Deletion of the N-terminal 11 aa (GtA_∆11aa) led to shorter (73.24 nm) and smaller filaments (19.49nm). In the case of GtD, filaments that had a length of 55.56 nm and diameter of 23.32 nm could be found, while the fusion of the GtA-derived 11 aa (GtD_(+)11aa(GtA)) changed the length of the filaments to 89.16 nm and the diameter to 23.07 nm on average were detected in the fraction 21. A larger size of filaments could also be observed for the fusion of GtE-derived 10 aa (GtD_(+)10aa(GtE), but the effect is less pronounced (65 nm length/23.8 nm diameter) as compared to GtD_(+)11aa(GtA). These data indicate that GtA-derived amino-terminal 11 aa strongly affect the filament size, especially a significant change the filament length is found (Table 2 and Figure 4C,D).

In addition, after sucrose gradient enrichment, the released viral particles were analyzed by qPCR and EM (Figure 4E–H). No significant difference in the number of released genomes and virion size could be observed between GtA, GtD and the variants (Figure 4E–G). In summary, the presence of 11 aa from GtA increases the size and the number of filaments, but has no impact on the number of released viral particles and viral particle size.

### 3.5. Deletion of 11 aa in the N-Terminal Part of GtA PreS1-Domain does not Affect the Viral Infectivity

The data described above indicate that neither the size nor the amount of released viral particles are affected in the variants. To study the potential effect on the infectivity, differentiated HepaRG cells were infected with supernatant derived from Huh7.5 cells expressing a genome, either the GtA or the GtA_∆11aa variant (Figure 5A–C). Monitoring the HBsAg and HBeAg (as an additional marker for HBV replication) in the culture supernatants by ELISA over a time of 11 days showed that comparable amounts of HBsAg and HBeAg were secreted from cells infected either by GtA or by GtA_∆11aa. In consistence with results from transient transfection experiments, a weaker LHB-specific signal was observed in the immunofluorescence staining for cells infected by GtA_∆11aa as compared to GtA. These data indicate that deletion of the N-terminal 11 aa of the PreS1 domain in GtA has no impact on viral infectivity. However, no productive infection was established in the case of HBV viral particles derived from cells overexpressing GtD_(+)10aa(GtE) and GtD_(+)11aa(GtA) neither for an infection with MOI = 200 nor with MOI = 1600 (Figure 5D,E). This indicates that with respect to the infectivity of the genotype D-derived variants, the GtD-specific backbone tolerates no changes in the N-terminal part of the PreS1-domain.

### 3.6. The 11 aa in the N-Terminal Part of GtA PreS1-Domain Harbor a Positive and a Negative Element Regulating LHB Amount and Subcellular Distribution

The loss of the N-terminal 11aa of LHBs GtA (GtA_∆11aa) is characterized by a lower amount of intracellular and extracellular LHBs. To refine the sequence requirements, a stepwise deletion within the 11 aa was performed by removal of aa 2–7, aa 3–8, aa 4–9, aa 5–10 and aa 6–11 (Figure 1). Moreover, two mutants (GtA_aa3–8(GtE) and GtA_aa7–12(GtE)) containing N- and C-terminal replacements derived from the corresponding region of GtE were generated (Figure 1). An analysis of these mutants revealed that N-terminal deletions within the region aa 2–9 (∆aa2–7, ∆aa3–8 and ∆aa4–9) increased drastically the amount of intracellular and extracellular LHBs, while C-terminal deletions within the region aa 5–11 (∆aa5–10, ∆aa6–11) decreased significantly the amount of intracellular LHBs. The amount of extracellular LHBs was not affected by these C-terminal deletions (Figure 6A–C). While N-terminal replacement (GtA_aa3–8(GtE)) showed no significant effects on the amount of intracellular and extracellular LHBs, the C-terminal substitution (GtA_aa7–12(GtE)) strongly reduced the amount of intracellular LHBs (Figure 6A–C). Like for GtA_∆11aa, almost no effects on the amounts of intracellular and extracellular core protein and HBsAg were observed if the N- and C-terminal deletions, excluding the GtA_∆aa6–11 variant, were expressed. In the case of the variant GtA_∆aa6–11 the amount of extracellular core was slightly but significantly reduced in the core ELISA (Figure 6D–H) Confocal immunofluorescence microscopy using the monoclonal antibody MA18/7 revealed that in cells overexpressing the mutant genomes with N-terminal deletions (∆aa2–7, ∆aa3–8 and ∆aa4–9) or substitution (aa3–8(GtE)), a significant fraction of LHBs was distributed in a GtA-like pattern with dot-like accumulation in the perinuclear region. However, it changed to the GtD-like pattern with diffuse staining in cells overexpressing the mutant genomes with C-terminal deletions (∆aa5–10, ∆aa6–11) or substitution (aa7–12(GtE)) (Figure 7). Taken together, these data indicate that the N-terminal part of these 11 aa acts as a negative element, while the C-terminal part acts as a positive element with respect to LHB amount. Detailed identification of the relevant aa residues triggering these effects is provided in Appendix A.

### 3.7. Deletion of 10 aa in the N-Terminal Part of GtE PreS1 Domain Increases the Production and Release of LHBs

As compared to GtA, the amino terminus of the PreS1 domain in the case of GtE contains completely different 10 aa. The question arises whether the GtE-derived 10 aa have the same function as GtA-derived 11 aa. To clarify this, deletions of aa 2–5, aa 2–8 and aa 2–11 were performed by shifting the methionine start codon to positions 5, 8 and 11 (Figure 1, W5M, P8M and W11M). In contrast to genotype A, the deletions of aa 2–5 and aa 2–8 decreased the amount of extracellular LHBs significantly, while the amount of intracellular LHBs was not affected considerably (Figure 8A–C). No significant change in the amount of extracellular HBsAg could be detected (Figure 8F). For unknown reasons, deletions of aa 2–5 and aa 2–8 caused a significant reduction of intracellular and extracellular core (Figure 8D,E). Deletion of aa 2–11 (W11M) strongly increased the amount of intracellular and extracellular LHBs (Figure 8A–C), while comparable amounts of intracellular and extracellular core and HBsAg could be detected as compared to GtE (Figure 8D–F), which implies that region aa 9–11 in the N-terminal PreS1 domain of GtE specifically exerts a negative effect on LHB production and release.

To study whether the effect of the N-terminal aa of the PreS1 domain of GtA or GtE on the amount of LHBs is limited to LHBs or exerts a more general effect, these sequences were fused to the amino terminus of eGFP (Figure 9A). The impact on the amount of GFP was analyzed by quantitative fluorescence microplate reader and by Western blotting (Figure 9B). Both approaches revealed that the GtE-derived N-terminal aa led to a much lower amount of the eGFP fusion proteins compared to the fusion protein harbouring the GtA-derived N-terminal aa. Analysis of the subcellular distribution of the fusion proteins revealed that the fusion protein harbouring the GtE-derived N-terminus homogenously distributed all over the cell including the nucleus comparable to eGFP. In contrast to this fusion of the GtA-derived N-terminal aa drastically changed the distribution of the resulting eGFP fusion protein to an extranuclear reticular distribution (Figure 9C,D).

This indicates that the effect of the GtA and GtE-derived N-terminal aa on the protein amount is not limited to LHBs. Apart from the effect on the amount the GtA-derived amino-terminal, 11aa have a strong impact on the subcellular distribution.

## 4. Discussion

Although the PreS1 domain plays a complex and vital role in the HBV life cycle, the isolates from chronic HBV carriers frequently contain in-frame deletions. Recently, variants with deletions in the PreS1 start codon associated with liver cirrhosis and hepatocellular carcinoma have been reported frequently from Asian patients infected by GtB and GtC [13,27,28,29]. Due to the strong impact of these deletions on the amount of LHBs, they could contribute to an escape from the immune response by lowering the amount of LHB-specific peptides presented on the cellular surface. Here, we describe GtA variants lacking the first methionine in the PreS1 domain that were identified in a European cohort of HBeAg negative patients. This indicates that deletions of the N-terminal part of the PreS1-domain are not restricted to GtC and GtB. No significant difference in serum HBsAg and HBV DNA could be observed for these variants of GtA. This is consistent with in vitro experiments; that the absence of these 11 aa has no impact on the levels of HBsAg and HBV DNA. However, the loss of N-terminal 11 aa strongly correlates with a lower level of intracellular LHBs. Further analyses revealed that this effect is sequence-dependent, as replacement of these N-terminal 11 aa in the PreS1-domain of GtA by the corresponding GtE-derived 10 aa (GtA_aa2–12(GtE)) fails to rescue the level of LHBs. Moreover, the impact on LHB amount is not caused by a difference in protein half-life, as the fusion of 11 aa to the N-terminus of the PreS1 domain of GtD does not significantly affect the half-life of LHBs (Appendix A). A detailed analysis of these amino-terminal 11 aa of GtA revealed the existence of distinct sequence elements affecting the amount of LHBs: a negative N-terminal element (aa 2–8) and a positive C-terminal element (aa 9–11). The data revealed that aa 9-11 play a predominant role in directing LHBs production, as it positively associates with the LHBs level independent from the presence or absence of the negative elements aa (2–8), All in all, the N-terminal 11 aa in the PreS1-domain of GtA correlate with LHBs yield, probably due to a more favorable context for efficient LHB initiation. This hypothesis is confirmed experimentally by the fusion of either the 11 aa of GtA or 10 aa of GtE to the N-terminus of eGFP. As compared to the fusion of GtE-derived 10 aa, the GtA-derived 11 aa lead to a significantly higher amount of the respective eGFP fusion protein. The presence of 11 aa not only leads to a higher amount of intracellular LHBs, but also causes a significant perinuclear accumulation of LHBs (GtA-like pattern). Interestingly the fusion of the GtA-derived 11aa to eGFP strongly affects the subcellular distribution of the resulting fusion protein. While eGFP or the fusion protein with the GtE-derived N-terminal amino acids is equally distributed all over the cell including the nucleus, the fusion protein harbouring the GtA-derived N-terminus displays a reticular distribution resembling an ER-like distribution, although the N-terminus does not represent a transmembrane region. An interaction with a protein on the cytoplasmic face of the ER membrane or an ER localization due to acylation might exist.

Apart from the effect on the amount and subcellular distribution presence of these GtA-derived 11a leads to longer LHB-containing filaments with a decreased release capacity. The decreased release might be caused by the structural changes that could be recognized as partial misfolding that impairs release.

Interestingly, the presence or absence of 11 aa only affects the size and number of filaments, but not viral particles. The fact that there is no difference in the morphology of viral particles might be explained by differences between the assembly of viral and filamentous particles, which could be partially supported by a previous publication describing that a PreS1 deletion mutant generates semi-enveloped viral particles and shorter filaments [26]. Moreover, there is no effect on the number of released viral particles. This seems to contradict to the previous studies that indicate that deletion of the PreS1 start codon increased HBV DNA level in chronically GtC-infected patients [15,30], however it could be also a combinatory effect of coexisting quasispecies in this patient group. Moreover, the difference may be because of the different genotypic background, i.e., genotype C versus genotype A. Furthermore, the 11 aa overlap with the spacer domain of polymerase that tolerates variable mutations without affecting polymerase function [31,32]. Our data are supported by the points that (i) the loss of 11 aa affects only the amount of LHBs, which is still high enough to enable the efficient production of viral particles and (ii) the amounts of HBsAg and core are not affected. This reflects that the limited amount of LHBs (GtA_∆11aa) is preferentially recruited for proper viral envelopment at the cost of assembling fewer and shorter filaments. An additional driving force for the formation of properly enveloped virions could be the LHB–capsid interaction that is absent in the case of filament formation.

Although a myristoylated peptide from LHBs containing the 11 aa with the point mutation K7T has been described to have slightly reduced binding capacity of the cellular receptor [33], our data show that the presence or absence of 11 aa in GtA has only a marginal impact on the viral infectivity in cell culture. This suggests that such a minor change in binding capacity of the PreS1-domain to the cellular receptor (complex) does not alter the viral infectivity significantly. However, the fact that N terminal insertions in the genotype D preS1 domain are not tolerated indicates that the backbone of the construct can affect the phenotype. It is well established that the PreS1 domain is acylated at glycine 2 with myristic acid and that myristoylation, in addition, depends on the amino acids at position 3, 6 and 7. This posttranslational modification (myristoylation) is indispensable for viral infectivity [34,35,36,37,38]. The first assumption would be that the loss of infectivity is due to the impaired myristoylation by insertions of 11 aa or 10 aa. Indeed, these insertions abolish the original myristoylation site in the N-terminus of the PreS1-domain as the methionine next to glycine is not removed. However, both the GtA-derived 11 aa or the GtE-derived 10 aa harbour a conserved N-myristoylation motif, which should be recognized by the N-myristoyl transferase. This is experimentally confirmed by a successful infection of differentiated HepaRG cells by virus produced from GtA_aa2–12(GtE) expressing cells (Figure 1 and Appendix A). Moreover, an in silico model predicts N-myristoylation (http://mendel.imp.ac.at/myristate/SUPLpredictor.htm). It can be hypothesized that the GtD PreS1/receptor recognition requires a highly conserved sequence/structure [33] depending on defined distances between crucial residues and thus tiny modifications around the receptor-binding domain abolish the interaction between PreS1 and the cellular receptor complex. A more distant fixation of the N-terminal part of the PreS1-domain in the membrane due to the myristoylation could lead to higher structural flexibility of the subsequent receptor binding domain that could affect the entry process.

Taken together, these data characterize, with respect to the different HBV genotypes, the N-terminal part of the PreS1 domain as a central factor affecting the amount and subcellular distribution of LHBs. Moreover, the release promoting and inhibiting aa residues within this region were identified and the general impact of these amino acids on the amount and subcellular distribution of proteins was demonstrated. With respect to the increasing number of HBV isolates from patients suffering from chronic HBV with mutations in the N-terminal part of the PreS1 domain, these data could contribute to a better understanding of viral persistence and virus-associated pathogenesis in these cases.

## Figures and Tables

**Figure 1 cells-09-01898-f001:**
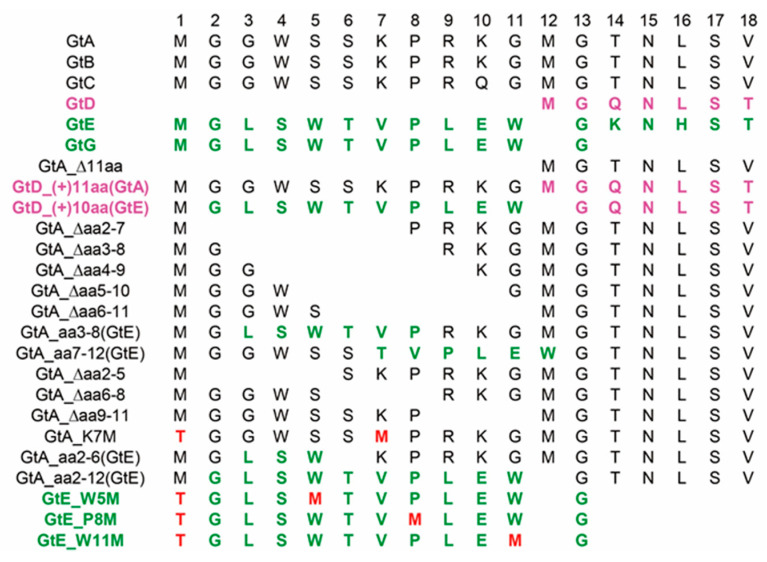
Sequence comparison of amino acids 1–18 in the pres1 of different genotypes and the derivatives.

**Figure 2 cells-09-01898-f002:**
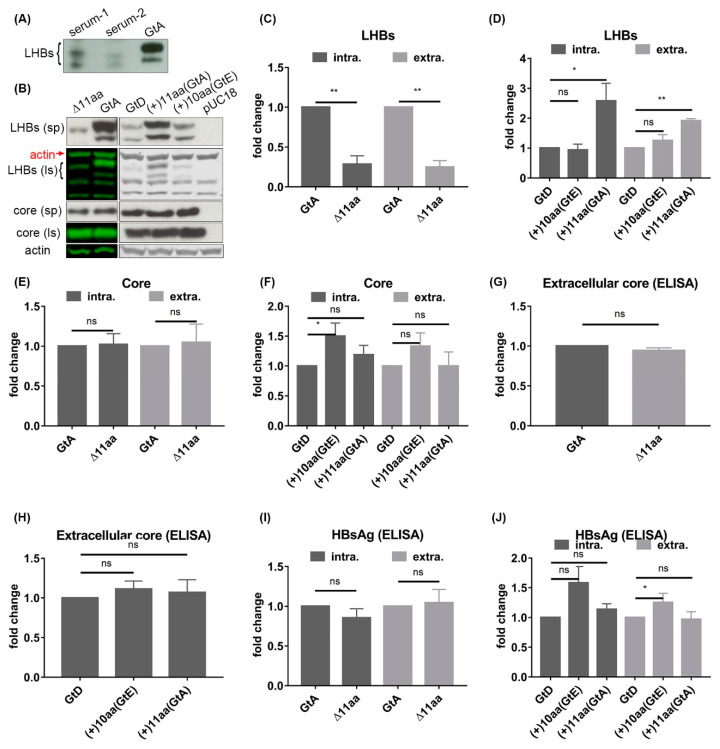
The presence of GtA-derived 11 aa is accompanied by higher amounts of intracellular and extracellular large surface proteins (LHBs). (**A**) Serums from patients infected with GtA variants containing PreS1 start codon deletion were analyzed by Western blot using a PreS1-specific serum (MA18/7). Purified wildtype GtA particles were used as control. (**B**) Supernatants and lysates from Huh7.5 cells transfected with genomes GtA, GtD and the derivatives were analyzed by Western blot using MA18/7 and a core-specific serum (K46). β-actin was used as loading control. (**C**–**F**) The signals of LHBs and core in the Western blot were quantified by Image studio lite from LI-COR Biosciences. (**G**,**H**) Supernatants from Huh7.5 cells transfected with genomes GtA, GtD and the derivatives were analyzed by core ELISA. (**I**,**J**) Supernatants and lysates from Huh7.5 cells transfected with genomes GtA, GtD and the derivatives were analyzed by HBsAg ELISA. The signal from GtA or GtD was standardized as 1. Ls, lysate; sp, supernatant. These quantitative data are mean values from three independent experiments. *, *p* < 0.05; **, *p* < 0.01.

**Figure 3 cells-09-01898-f003:**
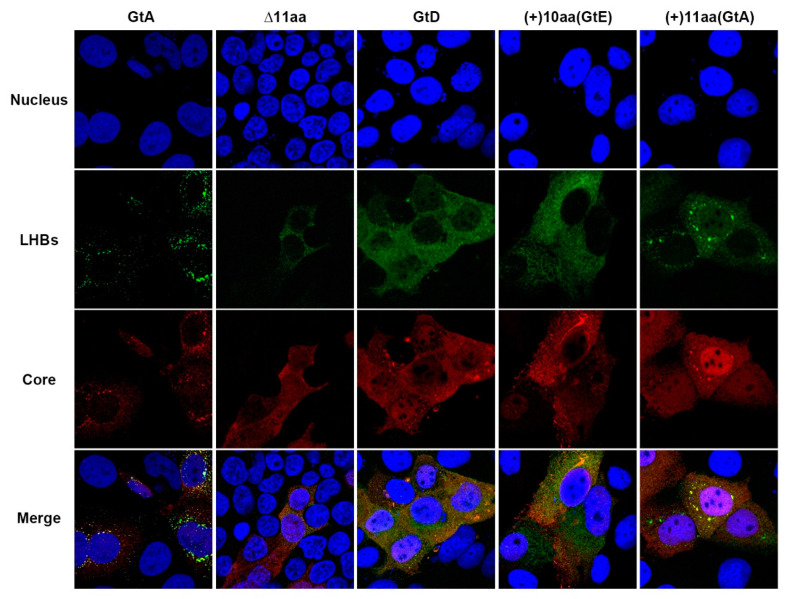
The presence of GtA-derived 11 aa causes significant subcellular accumulation of LHBs. Huh7.5 cells overexpressing genome GtA, GtD and the derivatives were analyzed by confocal laser scanning microscopy (CLSM) using a PreS1-specific monoclonal MA18/7 and a core-specific polyclonal antiserum (Dako). Nuclei were stained with DAPI (blue). The merge of the fluorescences were shown in the bottom row. The figures showed a representative result; comparable data were obtained for the analysis of 10 different cells.

**Figure 4 cells-09-01898-f004:**
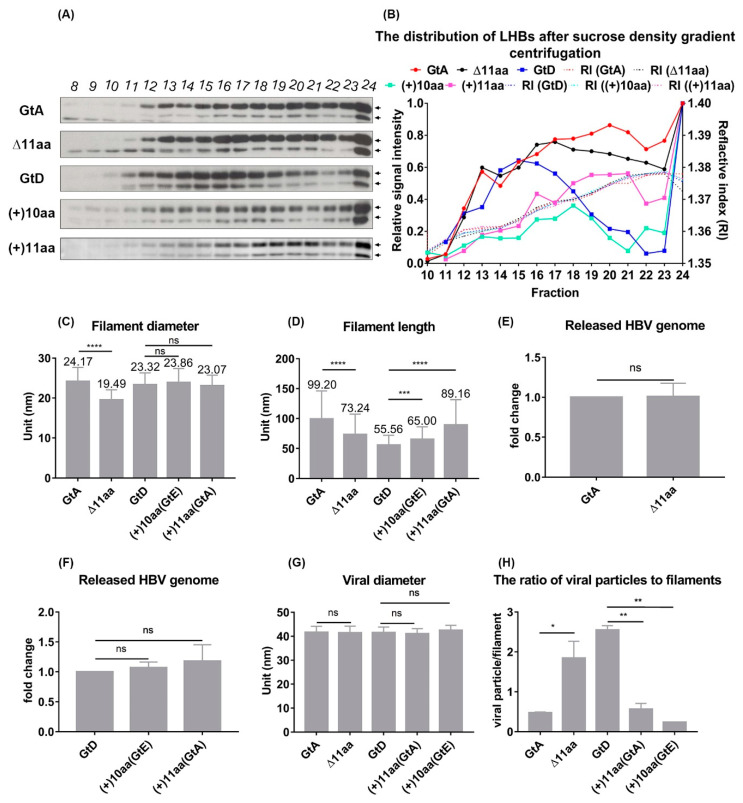
The presence of GtA-derived 11 aa promotes the generation of abundant longer filaments. (**A**,**B**) Polyethylene glycol (PEG)-concentrated supernatants from Huh7.5 cells overexpressing the genome GtA, GtD and the derivatives were analyzed by 10–30% (*w/w*) discontinuous sucrose gradient ultracentrifugation. A total of 24 fractions were harvested and fractions 10 to 24 were analyzed by refractive index, as well as Western blot using the PreS1-domain-specific monoclonal MA18/7. The signal of LHBs was measured by Image studio lite and the signal from fraction 24 was standardized as 1. (**C**,**D**) Filament-rich fractions (F20 in GtA, F17 in GtA_∆11aa, F15 in GtD, F21 in GtD_(+)11aa(GtA) and F18 in GtD_(+)10aa(GtE)) were analyzed by electron microscopy (EM). The size (diameter and length) of more than 80 filaments (101 in GtA, 136 in GtA_∆11aa, 87 in GtD, 108 in GtD_(+)11aa(GtA) and 81 in GtD_(+)10aa(GtE)) was measured by an EM-specific software Radius (EMSIS). (**E**–**H**) HBsAg-containing particles from Huh7.5 cells overexpressing GtA, GtD and the derivatives were precipitated by 10% PEG and then purified by sucrose ultracentrifugation followed by qPCR (**E**,**F**) and EM (**G**,**H**). (**G**) The diameter of more than 20 virions (47 in GtA, 92 in GtA_∆11aa, 90 in GtD, 42 in GtD_(+)11aa(GtA) and 22 in GtD_(+)10aa(GtE)) was measured by an EM-specific software Radius (EMSIS). *, *p* < 0.05; **, *p* < 0.01; ***, *p* < 0.001, **** *p* < 0.0001.

**Figure 5 cells-09-01898-f005:**
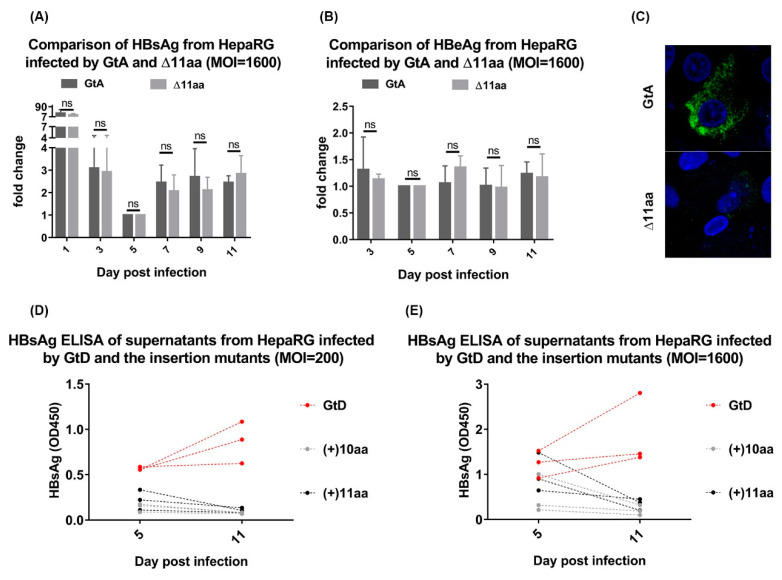
Deletion of 11 aa in the N-terminal part of GtA PreS1 has no impact on viral infectivity. (**A**–**C**) Differentiated HepaRG were infected with HBV from GtA and GtA_∆11aa at multiplexity of infection (MOI) 1600. Culture supernatants collected at indicated time points were analyzed by HBsAg ELISA (5A) and HBeAg ELISA (5B). The lowest signals (5 days post-infection) were standardized as 1. The cells were fixed and analyzed by CLSM using the MA18/7. (**D**,**E**) Differentiated HepaRG were infected with HBV from GtD, GtD_(+)10aa(GtE) and GtD_(+)11aa(GtA) at MOI 200 and 1600. Culture supernatants collected at 5- and 11-day post-infection were analyzed by HBsAg ELISA. These quantitative data are mean values from three independent experiments.

**Figure 6 cells-09-01898-f006:**
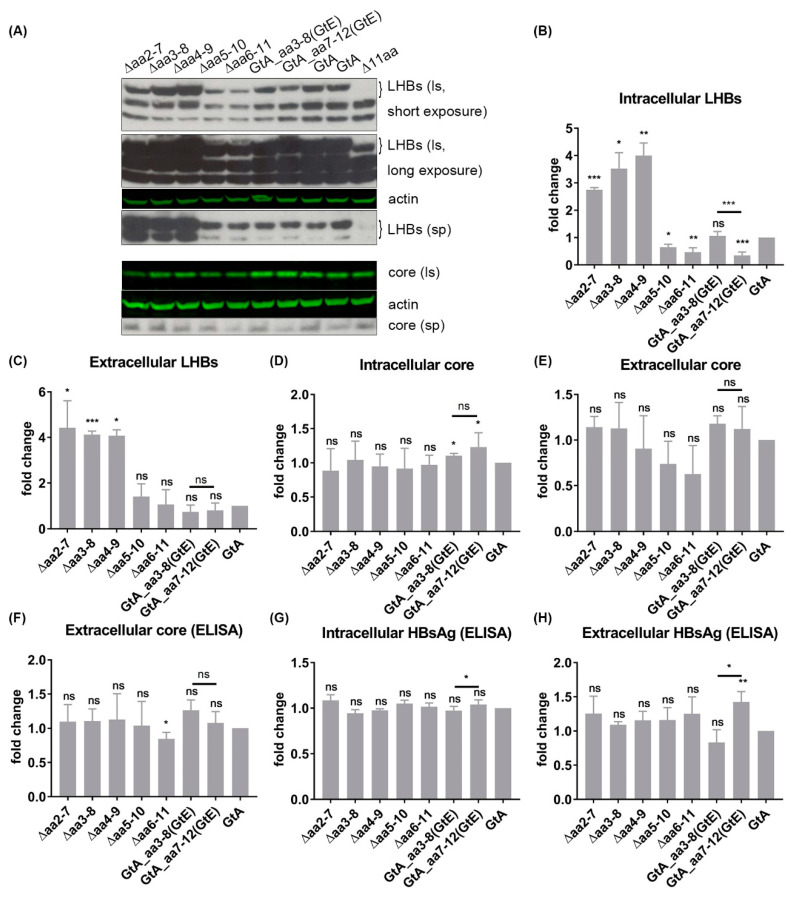
The GtA-derived 11 aa comprise a positive and a negative region regulating LHBs production and secretion. (**A**) Supernatants and lysates from Huh7.5 cells transfected with genomes GtA and the derivatives were analyzed by Western blot using a PreS1-specific serum (MA18/7) and a core-specific serum (K46). β-actin was used as loading control. (**B**–**E**) The signals of LHBs and core in the Western blot were quantified by Image studio lite from LI-COR Biosciences. (**F**) Supernatants from Huh7.5 cells transfected with genomes GtA and the derivatives were analyzed by core ELISA. (**G**,**H**) Supernatants and lysates from Huh7.5 cells transfected with genomes GtA and the derivatives were analyzed by HBsAg ELISA. The signal from GtA was standardized as 1. Ls, lysate; sp, supernatant. These quantitative data are mean values from three independent experiments. *, *p* < 0.05; **, *p* < 0.01; ***, *p* < 0.001.

**Figure 7 cells-09-01898-f007:**
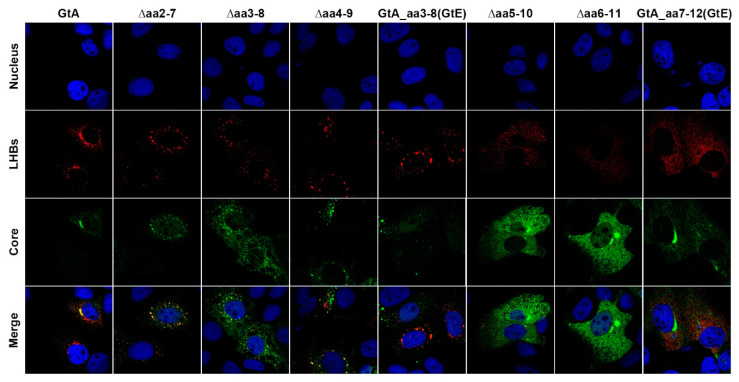
C-terminal deletions and substitution in 11 aa change the pattern of subcellular LHBs distribution. Huh7.5 cells overexpressing genome GtA and the derivatives were analyzed by confocal laser scanning microscopy (CLSM) using the LHB-specific monoclonal MA18/7 and core-specific polyclonal antiserum (Dako). Nuclei were stained with DAPI (blue). The merge of the fluorescences were shown in the right column. The figures showed a representative result; comparable data were obtained for the analysis of 10 different cells.

**Figure 8 cells-09-01898-f008:**
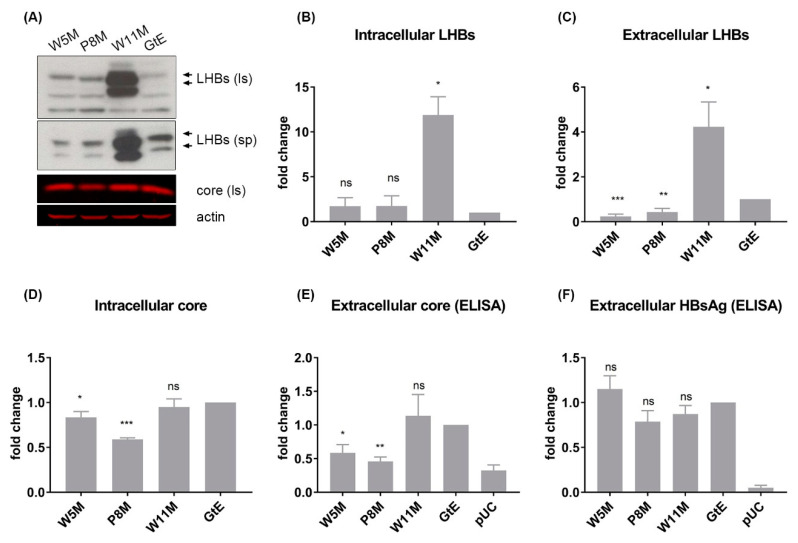
N-terminal 10-aa-deletion in GtE PreS1 domain increases dramatically the production and release of LHBs. (**A**) Supernatants and lysates from Huh7.5 cells transfected with genomes GtE and the derivatives were analyzed by Western blot using the PreS1-specific serum (MA18/7) and the core-specific serum (K46). β-actin was used as loading control. (**B**–**D**) The signals of LHBs and core in the Western blot were quantified by Image studio lite from LI-COR Biosciences. (**E**,**F**) Supernatants from Huh7.5 cells transfected with genomes GtE and the derivatives were analyzed by core ELISA and HBsAg ELISA. pUC18-transfected cells were used as negative control. The signal from GtE was standardized as 1. *, *p* < 0.05; **, *p* < 0.01; ***, *p* < 0.001.

**Figure 9 cells-09-01898-f009:**
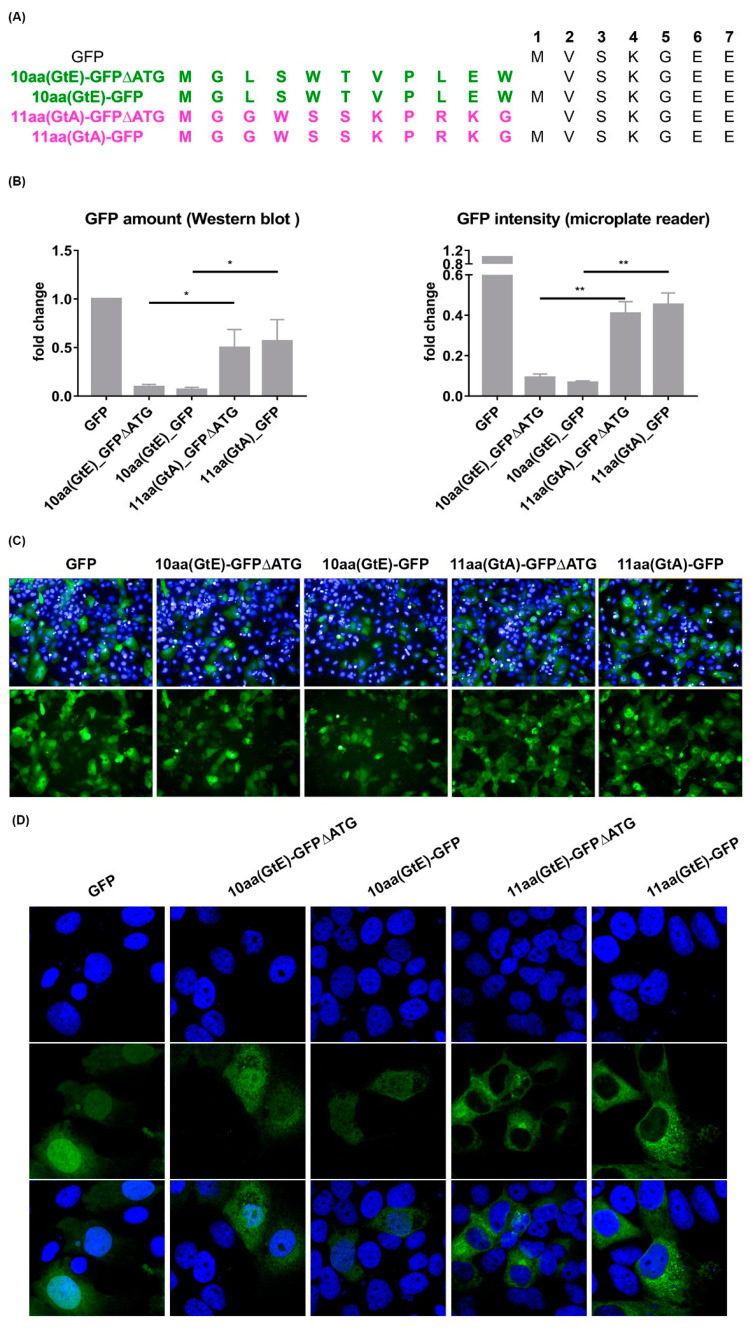
The impact of the N-terminal 11aa and 10aa on amount and distribution is independent from the LHBs context. (**A**) Schematic illustration of the start sites of eGFP and its fusion protein containing either 11 aa or 10 aa derived from N-terminal PreS1 of GtA or GtE. (**B**) eGFP level in Huh7.5 cells overexpressing genomes eGFP and its derivatives were analyzed by Western blot using an anti-eGFP specific serum and by microplate reader from Tecan by exciting the protein at 490 nm. As control, actin was used in Western blot, and in the microplate reader assay, nuclei were stained with DAPI and excited at 358 nm. (**C**) Huh7.5 cells overexpressing genomes eGFP and its derivatives were analyzed by Operetta high-content imaging system from PerkinElmer. Nuclei were stained with DAPI in blue. The signal from eGFP was standardized as 1. Ls, lysate; sp, supernatant. These quantitative data are mean values from three independent experiments. (**D**) CLSM analysis of Huh7.5 cells expressing these mutants. *, *p* < 0.05; **, *p* < 0.01.

**Table 1 cells-09-01898-t001:** Epidemiological, virological and serological characteristics of HBeAg negative patients infected chronically with hepatitis B virus (HBV) Genotype A.

Parameter	*n*	Age (Year, Mean ± SD)	Male Gender	HBV DNA (Mean log IU/mL ± SD)	qHBsAg (Mean log IU/mL ± SD)	ALT (Mean U/L ± SD)	Ethnicity		
							White	Asian	African-American
Total, *n* (%)	115	42.2 ± 12.5	44 (38)	2.7 ± 0.7	3.6 ± 0.9	29.1 ± 15.4	97	7	11
Deletions Covering the First Methionine	2	41.5 ± 5.5	0 (0)	2.6 ± 0.22	4 ± 0.9	24.5 ± 4.5	2	0	0

SD, standard deviation; qHBsAg, quantitative Hepatitis B surface Antigen; ALT, alanine transaminase.

**Table 2 cells-09-01898-t002:** Summary of the filament-representing fractions and filament size.

	GtA	∆11aa	GtD	(+)10aa	(+)11aa
Sucrose Fraction	F20	F17	F15	F18	F21
Diameter (nm)	24.1 ± 0.3	19.4 ± 0.2	23.3 ± 0.3	23.8 ± 0.3	23.0 ± 0.2
Length (nm)	99.2 ± 4.6	73.2 ± 2.9	55.5 ± 1.7	65.0 ± 2.0	89.1 ± 4.7

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
