# Peer review of "The N-Terminus Makes the Difference: Impact of Genotype-Specific Disparities in the N-Terminal Part of The Hepatitis B Virus Large Surface Protein on Morphogenesis of Viral and Subviral Particles"

_cells, 2020, doi:10.3390/cells9081898_

Round 1

Reviewer 1 Report

It is well known the importance of preS1 domain of the HBV L envelope protein. It is involved in receptor binding and viral entry, as well as in capsid binding and viral particle formation due to the dual topology of L protein.  In this manuscript, Bingfu Jiang et al, study the implication of the N-terminal deletions in the preS1 domain for LHBs, subviral and viral production and infectivity.

Major comments:

-Materials and methods are insufficiently described.

-Figure 2 shows a variation of intracellular core production, so that an analysis of viral transcripts is required, along with quantitative measurement of the viral genome. The same problem in fig. 6 and 8, the mutants were analyzed only at the protein level.

-The cellular distribution of wild-type and mutants LHBs are based on analysis of 10 different cells (fig 3). I don't think that's enough.

-In fig 4, was measured the fraction distribution only for L protein and not for S, which is the major component of the subviral particles. It is also useful to see the distribution of viral genome in all fractions not only in one and to show some electron microscopy picture to illustrate the difference between viral filaments.

-In Fig. 8, the authors state that the N-terminal 10-aa-deletion in the GtE PreS1 domain dramatically decreases LHB production, but in (A) and (B) an increased level of mutant L proteins is shown compared to the type wild (GtE).

-The control with eGFP (fig. 9) is convenient to study but not very well chosen, being localized in a different cellular compartment, not in ER, as LHBs.

-Jiang Li et al, addressed recently (Gut,2020) the same subject of preS1 deletions in HBV genotypes. The article was neither cited, nor discused.

Minor comments:

-use the full names of the people who gave you antibodies or plasmids

-too many colours on the graphics

Author Response

Response to reviewer 1

Reviewer 1:

-Materials and methods are insufficiently described.

We modified the materials and methods section and included additional information.

-Figure 2 shows a variation of intracellular core production, so that an analysis of viral transcripts is required, along with quantitative measurement of the viral genome. The same problem in fig. 6 and 8, the mutants were analyzed only at the protein level.

There is only a small change in the amount of core that could be caused by a variety of factors. In light of this we abstained to perform a quantification of the viral transcripts as other factors could be relevant as well.

-The cellular distribution of wild-type and mutants LHBs are based on analysis of 10 different cells (fig 3). I don't think that's enough.

In this point, we disagree with the reviewer as we do not observe a heterogenicity of the staining patterns.

-In fig 4, was measured the fraction distribution only for L protein and not for S, which is the major component of the subviral particles. It is also useful to see the distribution of viral genome in all fractions not only in one and to show some electron microscopy picture to illustrate the difference between viral filaments.

The manuscript focusses on LHBs. As shown in figure 2 there is only a minor effect on the amount of extracellular HBsAg (mainly representing spheres). In light of this we decided to concentrate on the LHBs data. Moreover we abstained to include the SHBs-data and genome data for clarity reasons to keep the graph especially figure 4B clear

-In Fig. 8, the authors state that the N-terminal 10-aa-deletion in the GtE PreS1 domain dramatically decreases LHB production, but in (A) and (B) an increased level of mutant L proteins is shown compared to the type wild (GtE). Typing error, change to increases

Thank you. This was indeed a typing error. We changed this.

-The control with eGFP (fig. 9) is convenient to study but not very well chosen, being localized in a different cellular compartment, not in ER, as LHBs.

We partially agree with the reviewer about this point. Using free eGFP that is not fused to a membrane protein gave us the opportunity to study the strong impact of the 11aa(GtA) on the localization of the respective fusion protein. Fusion of these 11 aa withdraws eGFP from the nuclear fraction and causes a perinuclear accumulation.

-Jiang Li et al, addressed recently (Gut,2020) the same subject of preS1 deletions in HBV genotypes. The article was neither cited, nor discused.

The preparation of our manuscript overlapped with the publication of this paper. We cite this paper in the discussion of the revised version..

Minor comments:

-use the full names of the people who gave you antibodies or plasmids

We changed this.

-too many colours on the graphics done

The graphs were changed.

Reviewer 2 Report

The manuscript by Jiang et al. "The N-terminus makes the difference: Impact of genotype-specific disparities in the N-terminal part of the Hepatitis B virus large surface protein on morphogenesis of viral and subviral particles" represents an interesting and novel work that identifies a novel region in the N-terminus of the large envelope protein of HBV (L), which regulates subcellular localization of the L and also influences the egress of the L from the cells expressing HBsAg. The work analyzed "extra" N terminal sequences from genotypes A and E as compared to the genotype D, which has shorter PreS1 domain. Using a number of the L mutants, the experiments compared the accumulation and egress of the L, HBsAg, and virions, and also evaluated how the N terminal "extra" sequences can affect the infectivity of HBV, when they are inserted into N-terminus of the L of genotype D or deleted from  N-terminus of the L of genotype A. The data are interesting, novel and contribute to our mechanistic understanding of (i) the regulatory properties of N-terminal sequences of the L, and (ii) differences in functional properties between different HDV genotypes that are important for HBV life cycle. Several mostly minor concerns were identified during the review and are described below.

Specific comments:

  1. It could be mentioned that in recent years it was shown that the cores that do not contain HBV rcDNA could be coated with the HBV envelope proteins and released.

  1. Current consensus is that there are nine HBV genotypes, A-I, and genotype J is still under debate, because it apparently represents a recombination event between two different genotypes.

  1. The N-terminus of the L bears myristoylation motif facilitating this host-mediated modification that is essential for HBV infectivity.

  1. It would be helpful to describe what ELISA kit was used for analysis of HBeAg.

  1. The materials and methods should include the isolation of rcDNA from serum and sequencing of the regions coding for HBsAg.

  1. The authors may want to include the discussion that the observed results for Gt(+)10aa (GtE) could be related to the fact there is M missing at pos +12 (Fig. 1).

  1. "Fusion of the GtA-derived 11 aa to the N-terminus of PreS1-domain in GtD switched the subcellular distribution of LHBs from the GtD pattern to the GtA-like pattern. " - In Fig. 3, it looks as a combination of GtD and GtA patterns for LHBs.

  1. It would be helpful to include additional information that HBeAg levels were used as a measure of HBV replication efficiency during experiments evaluating the infectivity of HBV virions coated with different versions of HBsAg (Fig. 5).

  1. It would strengthen the discussion if possible mechanism would be offered to explain why GtD did not tolerate the additions of 10 or 11 aa at the N-terminus, since such insertions seem to dramatically reduce the virus infectivity (Fig. 5).

  1. It looks like that either Fig. 5D or 5E should present HBeAg data, and Figure legend should mention HBeAg ELISA.

  1. Since it looks that HBeAg and HBsAg were measurable for mutated GtD variants with 10 and 11 aa insertions in infection experiments at day 5 post-infection, it would be helpful to discuss whether there was inefficient initial infection in these cases.

Author Response

Response to reviewer 2:

  1. It could be mentioned that in recent years it was shown that the cores that do not contain HBV rcDNA could be coated with the HBV envelope proteins and released.

This point is mentioned on page 2, line 50-51 of the revised manuscript and a corresponding reference is provided (Ning X, Luckenbaugh L, Liu K, Bruss V, Sureau C, Hu J. J Virol. 2018 Jun 29;92(14):e00272-18)

  1. Current consensus is that there are nine HBV genotypes, A-I, and genotype J is still under debate, because it apparently represents a recombination event between two different genotypes.

            This point has been modified (page 2, line 52)

  1. The N-terminus of the L bears myristoylation motif facilitating this host-mediated modification that is essential for HBV infectivity.

We addressed this relevant point (line 44) and included .Bruss V, Hagelstein J, Gerhardt E, Galle PR. Virology. 1996 Apr 15;218(2):396-9. as an additional reference.

  1. It would be helpful to describe what ELISA kit was used for analysis of HBeAg.

The HBeAg ELISA from SIEMENS, Germany was used. This information is included on page 3, line 88

  1. The materials and methods should include the isolation of rcDNA from serum and sequencing of the regions coding for HBsAg.

            This information is provided on page 3, lines 102-108

  1. The authors may want to include the discussion that the observed results for Gt(+)10aa (GtE) could be related to the fact there is M missing at pos +12 (Fig. 1).

 As requested by the reviewer we emphasized this point on page 2, lines 59-60.

  1. "Fusion of the GtA-derived 11 aa to the N-terminus of PreS1-domain in GtD switched the subcellular distribution of LHBs from the GtD pattern to the GtA-like pattern. " - In Fig. 3, it looks as a combination of GtD and GtA patterns for LHBs.

We modified this point (page 8, line 213)

  1. It would be helpful to include additional information that HBeAg levels were used as a measure of HBV replication efficiency during experiments evaluating the infectivity of HBV virions coated with different versions of HBsAg (Fig. 5). We included this information as requested by the reviewer (page  12, lines 276, 277)
  1. It would strengthen the discussion if possible mechanism would be offered to explain why GtD did not tolerate the additions of 10 or 11 aa at the N-terminus, since such insertions seem to dramatically reduce the virus infectivity (Fig. 5).

            This is included in the revised version on page 24, lines 456-476.

  1. It looks like that either Fig. 5D or 5E should present HBeAg data, and Figure legend should mention HBeAg ELISA.

Both graphs show data from HBsAg ELISAs, but the cells were infected with two different MOIs

  1. Since it looks that HBeAg and HBsAg were measurable for mutated GtD variants with 10 and 11 aa insertions in infection experiments at day 5 post-infection, it would be helpful to discuss whether there was inefficient initial infection in these cases.

We conclude from our data that the insertion of the 10 or 11 aa affect the infectivity of these mutants. Therefore, the HBsAg levels decrease and there is no increase as observed for the GtD wildtype. We clarify this point on page 24, lines 456-476.

Reviewer 3 Report

In their study Jiang and colleagues showed the following:

  • Addition of 11 aa present in genotype A, increases the amount of LHBs when inserted into a genotype D background.
  • The 11 aa when deleted from genotype A or inserted into a genotype D backbone affected the subcellular distribution of LHBs.
  • The sedimentation distribution of LHBs is affected by the presence or absence of 11 aa
  • The filament size of subviral particles differs in the presence of absence of 11 aa.
  • Infectivity of genotype A is not affected by the presence or absence of the 11 aa whereas genotype D does not tolerate the addition of either the genotype A 11 aa or the genotype E 10 aa.
  • The amino terminal of the 11 aa appears to have a negative regulatory role whereas the carboxyl end has a positive effect.

GENERAL COMMENTS

This is an interesting and comprehensive study. It is important to highlight that the effect of the 11 aa is in the context of subgenotype A2 and it may be automatically extrapolated to subgenotype A1, which has different molecular characteristics to subgenotype A2. A limitation of the study is that the equivalent experiments that were performed for subgenotype A2, were not performed for genotype E. In other words it would have been interesting to have included a comparison of genotype E, with and without the 10 aa.

In your introduction please explain why you chose to look at the effect of the 11 aa in a subgenotype A2 context instead of genotype B and C.

You have to highlight that differences in the phenotype can be influenced by the backbone of the construct. For example, genotype A is not identical to genotype D + 11 aa because there are other difference between genotype D and A. For example, genotype A has a 6 nucleotide insert at the carboxyl end of the core gene. This may not affect the expression of LHBs but it is important to be aware of these differences.

SPECIFIC COMMENTS

The title can be revised to:

The N-terminal of Hepatitis B Virus large surface protein makes the difference: Impact of genotype-type specific disparities on viral and subviral particle morphogenesis.

The manuscript should be edited by an English speaking individual for grammatical and language correction.

Lines 20/21: replace fusion with insertionLine 142

Line 25/28: replace motive with motif

Line 32: correct nomenclature family Hepadnaviridae (in italics)

Line 42: replace dpmain with domain, varies with is

Line 44: assemble “into” instead of to

Line 48: replace Besides with Furthermore

Line 50: It is now generally accepted that there are at least 9 genotypes, with a putative 10 genotype (please refer to Kramvis (2014) Intervirology for an update.

Line 61: This is not a recent finding. These mutations have been described in genotype A since the late 1990s and early 2000s!

Line 118: replace inoculums with inocula

Line 130/133: destruction is the incorrect term. Rather use deletion

Line 139, Figure 2A: What was the difference between serum 1 and serum 2? Did the HBV isolates belong to genotype A? Did the isolate in serum 2 have a deletion? Please clarify.

Line 142: any bioinformatics analysis and literature review would have revealed that these deletions are not restricted to genotypes B and C. Please cite a reference.

Line 153: The findings of you experiments cannot necessarily be extrapolated to genotypes B and C because the backbone of genotype B and C differs significantly from that genotype A and D and may result in a different outcome. Moreover your experiments only looked at subgenotype A2, which has different molecular characteristics to subgenotype A1, which influence the phenotype of the strain.

Line 154: Similarly genotype G differs from genotype E, especially in the core region where it has a 36 nt insert and thus the findings of your experiments cannot be extrapolated to genotype G.

Line 157: After domain add “,respectively.” Replace destruction with deletion

Figure 1: a limitation of the study is that you have not included a construct with a genotype E backbone with the 10 aa deleted equivalent to what you did for genotype A.

Line 165: replace fusion with insertion. Check throughout the manuscript.

Line 170: replace affects with affect

Line 199: decreased significantly – How was this measured? How many cells were quantified?

Line 207: Figure 3: figure legend. The merge is shown in the bottom row and not the right column! It seems that genotype D loses the localization in the secretory pathway because of the absence of the 11 aa. It would have been interesting to stain for the ER/ERGIC and to carry out colocalization studies. A negative control of only the secondary antibody should have been included to show specificity. 10 cells are not sufficient for statistical significance.

Line 259: Table 2: were the filament sizes significantly different?

Figure 5C: the cell chosen to show the distribution of the deletion mutant appears apoptotic. Please replace.

Line 354: replace smaller with lower

Figure 9C: what was the transfection efficiency for the different wells? Was it equivalent?

Line 366: 11 aa and 10 aa (10 aa missing)

Lines 423 – 423: Moreover, in the case……The difference may be because of the different genotypic background i.e. genotype C versus genotype A.

Line 440: the fact that N terminal insertions in the genotype D preS1 domain are not tolerated indicates that the backbone of the construct can affect the phenotype.

Authors must highlight the limitations of the study.  The backbone of the construct used can affect the phenotype of the expression of the different proteins.  So they must conclude that their observations are relevant for the backbone used in the experiments.  

Author Response

Response to reviewer 3:

In your introduction please explain why you chose to look at the effect of the 11 aa in a subgenotype A2 context instead of genotype B and C.

We addressed this point (lines 91-92)

The title can be revised to:

The N-terminal of Hepatitis B Virus large surface protein makes the difference: Impact of genotype-type specific disparities on viral and subviral particle morphogenesis.

We abstained from this as we think that the current title could be more clear. We hope the reviewer can accept our opinion.

The manuscript should be edited by an English speaking individual for grammatical and language correction.

 The manuscript has been revised by a native speaker

Lines 20/21: replace fusion with insertion, line 142

This has been changed.

Line 25/28: replace motive with motif

This has been changed.

Line 32: correct nomenclature family Hepadnaviridae (in italics)

This has been changed.

Line 42: replace dpmain with domain, varies with is

This has been changed.

Line 44: assemble “into” instead of to

This has been changed.

Line 50: It is now generally accepted that there are at least 9 genotypes, with a putative 10 genotype (please refer to Kramvis (2014) Intervirology for an update.

This has been changed.

Line 61: This is not a recent finding. These mutations have been described in genotype A since the late 1990s and early 2000s!

We changed this point as requested by the reviewer

Line 118: replace inoculums with inocula

This has been changed.

Line 130/133: destruction is the incorrect term. Rather use deletion

This has been changed.

Line 139, Figure 2A: What was the difference between serum 1 and serum 2? Did the HBV isolates belong to genotype A? Did the isolate in serum 2 have a deletion? Please clarify.

We clarified this point in the revised version of the manuscript.

Line 142: any bioinformatics analysis and literature review would have revealed that these deletions are not restricted to genotypes B and C. Please cite a reference.

We clarified this point and cited the reference describing the cohort (lines144-146)

Line 153: The findings of you experiments cannot necessarily be extrapolated to genotypes B and C because the backbone of genotype B and C differs significantly from that genotype A and D and may result in a different outcome. Moreover your experiments only looked at subgenotype A2, which has different molecular characteristics to subgenotype A1, which influence the phenotype of the strain.

We included a respective statement in the revised version of the manuscript.(line 171-174)

Line 154: Similarly genotype G differs from genotype E, especially in the core region where it has a 36 nt insert and thus the findings of your experiments cannot be extrapolated to genotype G.

We included a respective statement in the revised version of the manuscript.(line 171-174)

Line 157: After domain add “,respectively.” Replace destruction with deletion

This has been changed.

Figure 1: a limitation of the study is that you have not included a construct with a genotype E backbone with the 10 aa deleted equivalent to what you did for genotype A.

We agree with the reviewer, but we focused on the comparative analysis of A and D.

Line 165: replace fusion with insertion. Check throughout the manuscript.

We changed this.

Line 170: replace affects with affect

We changed this.

Line 199: decreased significantly – How was this measured? How many cells were quantified?

We focused on the impact on subcellular distribution and changed this point accordingly.

Line 207: Figure 3: figure legend. The merge is shown in the bottom row and not the right column! It seems that genotype D loses the localization in the secretory pathway because of the absence of the 11 aa. It would have been interesting to stain for the ER/ERGIC and to carry out colocalization studies. A negative control of only the secondary antibody should have been included to show specificity. 10 cells are not sufficient for statistical significance.

Line 259: Table 2: were the filament sizes significantly different?

The length of the filaments significantly differs (Figure 4C and D). We mention this in the revised version (line 253).

Figure 5C: the cell chosen to show the distribution of the deletion mutant appears apoptotic. Please replace.

We agree with the reviewer and changed this.

Line 354: replace smaller with lower

We changed this.

Figure 9C: what was the transfection efficiency for the different wells? Was it equivalent?

The transfection efficiency was about 30%. As the constructs are based on the same backbone, there is no difference between the different constructs with respect to the transfection efficiency, but different amounts of fusion proteins are produced.

Line 366: 11 aa and 10 aa (10 aa missing)

We changed this.

Lines 423 – 423: Moreover, in the case……The difference may be because of the different genotypic background i.e. genotype C versus genotype A.

We changed this.

Line 440: the fact that N terminal insertions in the genotype D preS1 domain are not tolerated indicates that the backbone of the construct can affect the phenotype.

We changed this.

Authors must highlight the limitations of the study.  The backbone of the construct used can affect the phenotype of the expression of the different proteins.  So they must conclude that their observations are relevant for the backbone used in the experiments.  

We agree with the reviewer and clarified this point (lines 171-174).

Round 2

Reviewer 1 Report

The authors did not  answered to my major concerns, they just „abstained to include the SHBs data and genome data for clarity reasons”. I think these data are very important for the conclusions of the article.